# Sliding Friction and Wear Characteristics of Wire Rope Contact with Sheave under Long-Distance Transmission Conditions

**DOI:** 10.3390/ma15207092

**Published:** 2022-10-12

**Authors:** Xiangdong Chang, Yuxing Peng, Zhencai Zhu, Hao Lu, Wei Tang, Xing Zhang

**Affiliations:** 1Jiangsu Key Laboratory of Mine Mechanical and Electrical Equipment, School of Mechanical and Electrical Engineering, China University of Mining and Technology, Xuzhou 221116, China; 2Jiangsu Collaborative Innovation Center of Intelligent Mining Equipment, Xuzhou 221116, China; 3Jiangsu Vocational Institute of Architectural Technology, School of Intelligent Manufacturing, Xuzhou 221116, China

**Keywords:** wire rope, sheave, bending contact, sliding friction, friction heat, surface wear, wear debris, tensile fracture

## Abstract

Wire rope has different degrees of surface wear under long-distance transmission conditions, which leads to performance degradation and greatly threatens its safety and reliability in service. In this paper, friction and wear tests between the transmission wire rope and sheave under different sliding velocities (from 0.8 m/s to 1.6 m/s) were carried out using a homemade test rig. The material of the steel wires was low carbon steel, and pulley material was ASTM A36 steel plate. The sliding friction coefficient (COF), friction temperature rise, wear characteristic parameters and wear mechanisms of the wire rope were analyzed. Additionally, the effect of different wear on the fracture behavior of the wire rope was investigated by a breaking tensile test. The results show that the average COF in the relatively stable stage decreased from approximately 0.58 to 0.51 with the increase of sliding velocity. The temperature rise of the wire rope increased rapidly with an increase of sliding velocity, from approximately 52.7 °C to 116.2 °C. The maximum wear width was the smallest when the sliding velocity was 1.2 m/s (approximately 1.5 mm). The surface wear was characterized by spalling, furrowing and plastic deformation, which are strongly affected by the sliding velocity. The wear mechanisms of the wire rope were mainly adhesive wear and abrasive wear. Surface wear changes the fracture morphology of the wire rope and accelerates its fracture speed.

## 1. Introduction

Wire rope is a key bearing component of hoisting, lifting, high altitude cableway and other driving force transmission and material transportation systems [1,2,3,4]. Its service status directly determines the safe and reliable operation of the equipment system and is closely related to the safety of the operator [5]. However, in practical engineering applications, there are friction and wear problems caused by different contact features of the wire rope, which greatly threatens the service safety and reliability of wire ropes [6,7]. In order to achieve long-distance and large-load material transmission, wire rope, in most cases, is used with different types of sheaves, as shown in Figure 1. The wire rope bends over the sheaves at different speeds and loads during the operation of transmission systems [8,9]. Thus, there is extrusion contact and relative sliding between the wire rope and the sheave groove, which causes surface wear of the wire rope and affects the stability of the system. Additionally, wire rope has a certain elasticity, which causes system vibration in the service process [10]. The sliding contact and surface wear of the wire rope is complex. Furthermore, wear degrades the performance of wire rope and accelerates its failure [11]. Therefore, it is necessary to study the sliding friction and wear behavior between the wire rope and the sheave, which is of great significance for the safe use of wire rope and prolongation of its service life.

In recent years, scholars have carried out a lot of research on damage and safe service of wire rope under actual working conditions. Wear is one of the main reasons of wire rope failure, and includes fretting wear between steel wires and surface wear of wire ropes [12,13]. Fretting wear behavior is mainly determined by different stress states and the service environment of the wire rope. Zhang et al. [14] studied the influence of different cross contact angles on fretting wear and fatigue damage of rope wires. The friction and wear coefficient decrease when the contact angle is small. Wang et al. [15,16] analyzed the fatigue damage and crack propagation characteristics of steel wires under different sliding parameters. Zhang et al. [17] investigated the effect of bending and dynamic tension on fretting fatigue behavior of rope wires. They found that the wear depth of steel wire decreases with the increase of sheave diameter. Additionally, the multi-wire spiral structure of wire rope leads to many different contact forms between its internal wires [18]. Based on Hertz contact theory and a semi-analytical method, Chen et al. [19,20] revealed the contact characteristics between the internal steel wires of wire rope under different loads. To explore the contact characteristics between the wire rope and the pulley, Hakala et al. [21] established a finite element model of contact between paste lubricated steel wire and cast iron. Furthermore, the service environment of the wire rope is very harsh [22]. The main wear mechanisms under different corrosive environmental media includes abrasive wear, adhesive wear, corrosion wear and fatigue wear [23,24]. Moreover, lubrication is an important way to protect the wire rope, and different lubrication states have great influence on friction and wear characteristics of wire ropes [25]. By adjusting the concentration of graphene and multilayer graphite additives, modified grease can improve the fretting wear behavior between steel wires and prolong the service life of the wire rope [26].

The surface wear of the wire rope is mainly caused by the contact collision between the wire rope and the extrusion contact between the wire rope and rope groove (drum and sheave) [27,28,29]. To study the friction and wear behavior between different wire ropes on the drum, Chang et al. [30] studied the tribological behavior of wire rope under different crossing angles and directions. The results show that cross direction had an obvious effect on the distribution of wear scar and wear morphology of wire rope. Oksanen et al. [31] investigated the wear mechanism of relative sliding between roller and wire rope under different contact conditions, and found spalling is the main mechanism of material removal on the wear surface. Considering the influence of hoisting system vibration, Zhang et al. [32] carried out friction and wear tests between wire ropes under different longitudinal vibration amplitudes and frequencies. Additionally, different working conditions affect the wear evolution characteristics of wire rope, especially corrosive solutions, which are most harmful to the friction and wear of wire rope [33]. Furthermore, surface lubrication can effectively reduce the infringement of a corrosive environment on the wire rope [34,35]. However, the open-air service conditions of wire ropes can easily lead to performance degradation and failure of lubricating oil. Therefore, the wear problem of wire rope is inevitable. A comprehensive understanding of the tribological properties of the wire rope is an important prerequisite to ensure its safe and reliable service.

There has been much previous research on the friction and wear of wire ropes, but these studies mainly focused on fretting wear between internal rope wires. The surface wear characteristics of wire rope is closely related to different contact forms. Existing research focuses on sliding friction and wear behavior between two wire ropes, which is different from the actual working conditions, while research on surface wear of wire rope caused by the contact between the wire rope and the sheave has rarely been carried out. Although rolling contact is the main contact form between the wire rope and the sheave during the operation of a transmission system, friction and wear caused by the sliding contact is a more serious problem. Moreover, when the frequency of use of the wire rope is high, wear and deterioration are a continuous processes during long-term service. Therefore, it is of great engineering significance to study the sliding friction characteristics between the wire rope and the sheave to ensure the safe service of wire ropes and reduce damage.

The aim of this paper was to reveal the friction characteristics and wear mechanisms of wire rope in sliding contact with the sheave. A rope-sheave friction testing machine was developed and sliding friction tests at different sliding velocities were carried out using a homemade test rig. Using the data acquisition system of the test rig and an infrared thermal imager, variations of the COF and friction temperature rise of the wire rope under different sliding conditions were revealed. Additionally, distribution of wear scar on the rope strand was analyzed. The wear width and wear amount of rope samples under different test conditions were measured and analyzed. Furthermore, the wear debris characteristics, wear morphology and wear mechanism of wire rope were analyzed by an optical microscope and SEM (scanning electron microscope) (Mocai Materials Science and Technology Co., Let., Shanghai, China). Finally, the influence of surface wear caused by rope-sheave contact on the fracture characteristics of rope wires was analyzed using a breaking tensile test.

## 2. Materials and Methods

### 2.1. Test Sample

The friction pair of the sliding friction test in this paper was composed of a wire rope and sheave groove. The surface structure is shown in Figure 2. The wire rope was composed of six spiral strands and one fibre core. Each strand contained 19 galvanized steel wires. The chemical composition of the wire material is presented in Table 1. Additionally, the detailed structure parameters are presented in Table 2.

The sheave was custom-made from ASTM A36 steel plate. The diameters of the sheave and the rope groove were 400 mm and 14 mm, respectively. The chemical composition of steel plate material (in wt%) was 0.18 C, 0.46 Mn, 0.275 Si, 0.045 S, 0.04 P.

### 2.2. Test Procedure

The friction and wear test rig of the wire rope and sheave is shown in Figure 3a.

The system included a driving device, loading device, sliding bracket and data acquisition system. The variable frequency motor drives the sheave to create continuous rotation, and there is a dynamic torque sensor connected between them. The floating loading bracket is connected to four linear guides mounted on the test stand frame through eight sliders. Additionally, both ends of the rope sample are made into rope buckles and fixed on both sides of the floating loading bracket, as shown in Figure 3b. Thus, under the action of gravity, the wire rope contacts with the sheave groove, as shown in Figure 3c. The contact load between the wire rope and the sheave is the sum of gravity of the floating load support and the wire rope. The sliding velocity of friction experiment is realized by adjusting the rotational speed of the sheave. The frictional drag torque and frictional heat of the test process can be monitored by the dynamic torque sensors and an infrared thermal imager. Furthermore, due to the structure of the sheave groove being relatively regular, it hardly changes much during the friction test. The wire ropes were replaced in different tests and the rope grooves were cleaned with alcohol after each test. To reduce the influence of test error, each test was repeated three times and the final results were averaged. In engineering applications, the types of sheaves used in conjunction with wire ropes are diverse, and the contact angle between the wire rope and the sheave varies from 0° to 180°. Additionally, the sliding speed and contact load between the wire rope and the sheave are constantly changing during operation. Therefore, we chose the sliding velocity as the research variable, and the selected test parameters were real under service conditions. Detailed parameters of the sliding friction and wear test are presented in Table 3. 

To explore the fracture behavior of a worn wire rope under tensile load, a breaking tensile test was carried out for the wire rope samples after the wear test. The tensile testing machine is shown in the Figure 3d. It was used to provide axial load for the rope samples. After the worn rope strand broke, the test was stopped immediately.

### 2.3. Test Parameters and Methods

The main objects and parameters of this study are shown in Figure 4.

During the sliding friction test with monitoring of the friction torque in real time by the torque sensor, the diameter of the sheave and the contact load can be used to calculate the sliding COF of the wire rope. Then, a curve of the COF with the sliding distance can be obtained, as shown in Figure 4a. Additionally, the maximum temperature of the sliding contact area of the wire rope can be monitored and recorded by an infrared thermal imager (see Figure 4b). Therefore, combined with the room temperature data, the curves of friction temperature increase of the wire rope under different sliding velocities can be obtained. After the friction test, the wear debris was collected and the wear loss of the friction pair was obtained using an electronic analytical balance (see Figure 4c). The measurement accuracy of the analytical balance was 0.1 mg. Each test was done three times. Furthermore, the distribution of surface wear scar on the rope sample and the maximum wear width of the rope strand were studied using an optical microscope (see Figure 4e,f). The surface wear characteristics of the wire rope and the fracture mechanism of the damaged rope wires were analyzed by the SEM (see Figure 4g,d).

## 3. Results and Discussion

### 3.1. Friction Parameters

Curves of the COF under different sliding distance and sliding velocities are shown in Figure 5a,b respectively.

With the increase of sliding distance between the wire rope and the sheave, the variation of COF under different sliding conditions was very obvious (see Figure 5a). At the initial stage of the sliding test, the COF rose rapidly to its maximum and then decreased rapidly. This behavior is closely related to the structure of wire ropes. The surface of the wire rope is a discontinuous structure consisting of multiple steel wires and strands. Thus, the contact form between the wire rope and the sheave groove are mainly the point contact and line contact at the beginning of the sliding friction test. Additionally, the contact stress of the friction pair is large under this condition, and the steel wire on the surface of the wire rope has a certain spiral angle, resulting in a large sliding friction resistance. Then, with the increase of sliding distance, the contact area between the wire rope and the groove becomes larger with increasing surface wear. Then, the contact form of the friction pair becomes surface contact, and the contact stress decreases continuously. Thus, the COF of the wire rope shows a decreasing trend in this process. Furthermore, when the sliding distance increases from approximately 50 m to 240 m, the COF decreases slowly at first and then gradually becomes stable. This means the contact area between the wire rope and the groove does not change much during this sliding friction and wear process, and the contact surface enters a relatively stable state. Because the wire rope is a cylindrical structure, the contact area does not always increase during the sliding wear process. With the removal of the surface material, the sliding contact area increases rapidly at first then the rate slows down and gradually reaches the maximum, which is consistent with the variation of the COF.

To quantitatively analyze the influence of different sliding velocities on the COF of the wire rope, data of the relative stable stage of different friction curves were calculated. The obtained average COF values are presented in Figure 5b. When the sliding velocity increases from 0.8 m/s to 1.6 m/s, the COF decreases approximately linearly, from approximately 0.58 to approximately 0.51. This indicates that the surface of the friction pair is smoother under the condition of high-speed sliding. The faster the sliding velocity between the wire rope and the rope groove, the faster the friction heat generated, and temperature of the contact area increases. Thus, the friction surface material softens and is easier to remove. Additionally, the sliding contact surface of the wire rope and the rope groove is more stable and not easy to change under high-speed sliding condition. This results in a smoother and more complete contact surface, which is an important reason for the reduced COF. Therefore, in practical engineering applications, the friction resistance between the wire rope and the sheave can be controlled by adjusting the running speed of the wire rope. This can better guarantee the service stability of the long-distance transmission system.

Friction heat is another important parameter that changes during sliding friction and wear. Figure 6 shows infrared thermography of the frictional contact area between the wire rope and the sheave under different sliding conditions.

Figure 6a is physical comparison of the contact structure of the friction pair. Figure 6b–f shows the infrared thermograms of contact pairs at different sliding velocities before the end of each sliding friction test. It can be seen that the color of the rope-sheave contact part is brighter. This means that the temperature of the wire rope is high, and the temperature rise is obvious in the friction test. Additionally, the surface color of the wire rope gradually darkens along the groove contact direction. The top of the contact area between the wire rope and the sheave is the brightest. Closer to the fixed end, there is a darker surface color. This means that the temperature distribution on the surface of the wire rope in the contact area is uneven. This is closely related to the stress distribution in the arc contact area of the wire rope. It is difficult to distinguish the effect of different sliding velocities on friction heat by infrared thermography. But it can be preliminarily judged by the change of scale that the surface temperature of wire rope increases with the increase of sliding velocity.

Infrared thermal images of the wire rope under different sliding distances are presented in Figure 6g–l. It can be seen that the color of the wire rope changes obviously during the sliding friction test. This means that the surface temperature of the wire rope increases with increasing sliding distance. At the beginning of the test (10 m), the temperature of the upper rope surface is low, and the friction heat is concentrated on the contact surface. As the sliding distance continues to increase, the surface temperature becomes increasingly higher. Especially in the first 50 m, the change of the surface color is very obvious. Furthermore, the surface color of the wire rope is bright and evenly distributed when the sliding distance exceeds 100 m. This indicates that the surface temperature of the wire rope is higher. The friction heat generation and heat dissipation enter a relative balance state. This indicates that the surface temperature of the wire rope is relatively high and stable. Therefore, through the change of infrared thermal image, it can be found that the surface temperature of the wire rope increases in the process of sliding friction, but the increase is not uniform.

To quantitatively analyze variation of the frictional heat under different sliding conditions, the maximum temperature increase on the surface of contact area of the wire rope was calculated, then the temperature rise curves under different sliding velocities were obtained, as shown in Figure 7.

The results are different from the curves of COF. In the first half of the friction test (approximately from 0 m to 100 m), the friction temperature rise increases rapidly. Then, it gradually becomes stable in the second half of the test. Additionally, when the sliding distance is less than 50 m, the temperature curves under different sliding velocities fluctuate greatly, as shown in Figure 7a. This is because the surface temperature of the wire rope changes rapidly. The frictional heat is transferred from the contact surface to the outer surface, and the highest temperature point is not fixed at the beginning of the friction test. Furthermore, the sliding friction between the wire rope and the sheave does not cause the surface temperature of the wire rope to rise continuously, and the temperature rise remains basically unchanged after the sliding distance exceeds 150 m. As the heat dissipation condition of the wire rope is constant, this indicates that the friction and wear of the wire rope has reached a relative equilibrium state. Furthermore, Figure 7b shows the average temperature rise of the wire rope in a relatively stable stage. It is clear that the friction temperature rise rapidly increases from approximately 52.7 °C to 116.2 °C with increase of sliding velocity. This means that the faster the sliding velocity between the rope and the groove, the more friction heat is generated and is more likely to cause the surface temperature to rise. Moreover, when the velocity increases from 1.2 m/s to 1.6 m/s, the rate of temperature rise slows down obviously. This is because the increase of sliding velocity accelerates the heat dissipation speed of the rope surface. Although the heat generated by friction increases rapidly, the increase of temperature is small. Therefore, the sliding velocity has a significant effect on the friction temperature rise between the wire rope and the sheave. The friction and wear state of the friction pair can be judged by the change of the temperature in the contact area.

### 3.2. Characteristic Parameters of Surface Wear Scar

Figure 8 shows the distribution of wear scar on the rope surface. It can be seen that the wear scar is composed of many wear surfaces on the strands.

Because of the arc contact between the wire rope and the sheave groove, the stress distribution of the whole contact surface is uneven. The characteristics of the wear scar on the rope strand vary greatly at different contact positions, as shown in Figure 8a. At both ends of the wear scar, the surface wear of the wire rope is relatively mild and the wear area on the strand is smaller, as shown in Figure 8b,g. As the contact position moves to the middle, the wear scar on the strand surface increases significantly (see Figure 8c,f). In the middle of the wear scar, the surface wear of wire rope is the most serious. Additionally, the wear surface is more complete and the characteristics of plastic deformation is more obvious under this condition. Furthermore, this characteristic of the wear scar on the rope surface is directly related to the rope-sheave contact characteristics. The contact stress on the rope surface decreases from the top to both sides of the contact arc. Therefore, the wear scar of the wire rope caused by sliding between the wire rope and the sheave is most obvious in the middle of the contact region. The wear area is the largest and it is continuously reduced towards both ends of the sliding contact surface.

To compare the effects of different sliding velocities on the surface wear of wire rope, the wear surfaces on the rope stand in the middle area of the wear scar under different velocities are presented in Figure 9.

It can be seen that the wear on the rope strand is distributed on the upper surface in an oval shape. Because the strand and steel wire are both cylindrical structures, the surface wear scar on the strand is arc-shaped and on the surface of the steel wire is oval. Thus, the contour curve of the wear scar is very irregular. Additionally, when the sliding velocity increases from 0.8 m/s to 1.2 m/s, the wear scar is significantly reduced, and the fluctuation of the profile curve is greater (see Figure 9f). This indicates that the increase of sliding velocity effectively protects the surface of wire rope and reduces its wear. However, as the sliding velocity increases from 1.2 m/s to 1.6 m/s, the wear scars become larger, and the surface is very smooth (see Figure 9d,e). This means that the influence of sliding velocity on the surface wear of the wire rope is variable. The sliding speed mainly affects the temperature of the contact surface. The increase of temperature leads to the softening of the friction pair material and the sliding surface contacts more fully. Furthermore, the wear debris is more easily deformed during the extrusion process, and the ploughing effect of the abrasive particles is reduced. This helps to slow down the removal of rope surface material. Moreover, when the sliding speed increases to a certain extent, the temperature of the sliding surface becomes very high, and a large amount of wear debris is removed by high-speed sliding. The characteristics of adhesion and plastic deformation between the friction pairs become more severe. Thus, the wear scars of the wire rope become larger under this sliding conditions.

Figure 10 shows the wear characteristic parameters of the wire rope under different sliding velocities.

The maximum wear width of the wire rope obtained from Figure 9 is presented in Figure 10a. This allows quantitative analysis of the effect of different sliding velocities on the wear characteristics of rope samples. When the sliding velocity increases from 0.8 m/s to 1.2 m/s, the wear width decreases from approximately 1.94 mm to approximately 1.5 mm, then it rapidly increases to 2.16 mm as the velocity continues to increase to 1.6 m/s. Therefore, under the condition of sliding wear between the wire rope and the sheave, the surface wear degree of wire rope decreases first and then increases with the increase of sliding velocity. In an engineering application, it is necessary to accurately control the running speed of wire rope to realize the effective protection and reduce the wear of wire ropes. Figure 10b shows the wear loss of the friction pairs. It can be seen that the wear loss decreases from approximately 5.614 g to approximately 3.991 g as the sliding velocity increases. Additionally, when the sliding velocity is less than 1.2 m/s, the wear loss changes slightly, by about 0.369 g. However, when the sliding velocity continues to increase to 1.4 m/s and 1.6 m/s, the wear loss decreases rapidly, and the variation is approximately 1.3 g. This indicates that a large amount of wear debris is produced in the process of sliding wear, which leads to variation of wear loss at different velocities not consistent with the variation of the wear width. Moreover, the increase of velocity effectively reduces the material removal of friction pairs. Although the wear area on the surface of the wire rope increases, the wear on the surface of the groove is reduced. Therefore, the total wear loss decreases with increasing sliding velocity.

### 3.3. Wear Mechanism

Figure 11 shows the SEM of the wear surface under different sliding velocities. When the sliding velocity is small, the wear surface of wire ropes is rough and the furrows generated during the sliding wear are very obvious, as shown in Figure 11a,b.

This indicates that the surface wear of the wire rope is very severe. Additionally, there are obvious peeling pits and spalling layers on the wear surface. This means that the adhesion of the sliding wear between the wire rope and the groove is serious, and the size of the wear debris is large under this sliding conditions. Furthermore, the wear surface is obviously smoother when the sliding speed increases to 1.2 m/s (see Figure 11c). The pits become smaller and there is obvious plastic deformation at the edge of the wear surface. This means that the size of the wear debris produced during the sliding wear test is significantly reduced. Thus, abrasive wear is not obvious, and the furrows become small on the wear surface. This effectively slows down the removal rate of the surface material of the wire rope. Because of the high surface temperature of the wire rope, the material is prone to plastic deformation during the sliding extrusion process. As a result, the wear is relatively mild, and the surface is smoother. Moreover, when the sliding velocity increases to 1.4 m/s and 1.6 m/s, it is difficult to see furrows and spalling pits on the wear surface (see Figure 11d,e). This means that the wear surface of the wire rope is very smooth, and the frictional resistance is small, which is consistent with the variation of the COF in Figure 5. Combined with the results of friction temperature rise (see Figure 7), it can be seen that the surface of sliding friction between the wire rope and the sheave is smoother and more regular under high temperature conditions. Therefore, the wear characteristics of the transmission wire rope mainly include furrow, pitting, spalling and plastic deformation. However, there are differences in feature types at different sliding velocities. The increase in sliding velocity causes the furrow to become smaller and the plastic deformation to become more obvious. The wear mechanisms are abrasive wear and adhesive wear, and the adhesion is weakened under high-speed sliding condition. 

Figure 12 shows the morphology of the wear debris produced in the friction and wear tests.

At the beginning of the test, the contact stress between the wire rope and the rope groove is large. The spiral steel wires on the surface of the wire rope have an obvious ploughing effect on the rope groove and produce long strip wear debris with larger size, as shown in Figure 12a–c. Additionally, most of these wear debris is generated from the surface of the rope groove. The ploughing action between the friction pairs is obviously greater than the adhesion action under this test conditions. Furthermore, as the sliding distance increases, the sliding contact surface goes through the running-in stage, and the wear debris size becomes significantly smaller, as shown in Figure 12d–f. This indicates that the ploughing effect of the contact surface between the wire rope and the groove is weakened, and the adhesive wear plays a leading role. The surface of the wire rope becomes smoother under this sliding condition. Moreover, Figure 12g–i shows the SEM micrographs of the wear debris under different multiples. It can be seen that the wear debris is irregular flake particles. There is obvious plastic deformation and cracks on the surface. This means that much wear debris is squeezed in the wear contact area and plays the role of ploughing and lubrication in sliding friction and wear. Therefore, the characteristics of the wear debris generated by the wire rope and the sheave during the relative sliding process are constantly changing, which is related to the friction characteristics and wear mechanism of the wire rope. It is helpful to judge the wear state of the wire rope by analyzing the characteristics of the wear debris.

Wire rope is a mechanical component with independent bearing capacity. In order to explore the influence of surface wear on the fracture characteristics of wire rope, the fracture morphology of steel wire for different rope samples was analyzed, as shown in Figure 13.

When the rope wire is undamaged, the fracture morphology of steel wire is obvious (see Figure 13a). The diameter shrinkage, fibre zone and shear lip characteristics left during the fracture process can be seen clearly. Additionally, there are small dimples in the fibre zone. This indicates that the stress of the wire section is relatively uniform. The fracture characteristics are regular and complete. Thus, the steel wire undergoes large plastic deformation and apparent yielding before fracture. Furthermore, when the surface of the steel wire suffers wear, the diameter shrinkage at the break region of the wire is no longer apparent and it is difficult to distinguish between the fibre zone and the shear lip characteristics (see Figure 13b). This indicates that the rope wire breaks faster. There is an incomplete steel wire section resulting in uneven stress distribution, but the steel wire maintains a large carrying capacity. The size of the dimples on the fracture surface is still small and evenly distributed. Moreover, when the surface wear of the rope wire is severe, the fracture characteristics of the wire become very obvious (see Figure 13c). Only an irregular fibre zone can be seen on the fracture surface. Additionally, the size of dimples on fracture surface increases significantly and varies in size. This means that the fracture speed of the rope wire is very fast. A sudden fracture occurs with almost no plastic deformation, resulting in a significant reduction in the bearing capacity of the wire. Therefore, wear may affect the fracture failure process and bearing capacity of the wire rope. With the increase of wear degree of wire rope, the plastic deformation of the rope under tensile load decreases, and the fracture speed increases. In engineering applications, surface wear can lead to local uneven stress distribution and is more likely to induce a sudden fracture of the wire rope.

## 4. Conclusions

In this research, the tribological behavior between a transmission wire rope and the sheave under different sliding velocities was revealed by experimental study. The major conclusions are as follows:(1)The COF between the wire rope and the sheave groove is constantly changing in the process of sliding friction. It decreases rapidly and then gradually stabilizes with the increase of sliding distance. In the relatively stable stage, the average COF decreases from approximately 0.58 to 0.51 with the velocity increases from 0.8 m/s to 1.6 m/s.(2)In the process of sliding friction, the friction heat generated between the friction pairs is a process of continuous accumulation. The surface temperature of the wire rope increases rapidly at first and then gradually stabilizes. The maximum temperature rise of the wire rope increases approximately linearly from 52.7 °C to 116.2 °C with increasing sliding velocity.(3)In the arc contact area, the surface wear at both ends of the wire rope is mild and the most severe surface wear occurs in the middle position. The maximum wear width of the wire rope decreases first and then increases with the increase of sliding velocity. It is the smallest when the sliding velocity is 1.2 m/s, which is about 1.5 mm.(4)The surface wear characteristics of the transmission wire rope are spalling, furrow and plastic deformation. The wear mechanisms are mainly abrasive wear and adhesive wear. The increase of sliding velocity weakens the ploughing effect and enhances adhesion.(5)Surface wear reduces fracture morphology and accelerates fracture speed of the wire rope. The fracture mechanism of the wire rope under tensile load is ductile fracture and severe wear leads to increased dimple size.

## Figures and Tables

**Figure 1 materials-15-07092-f001:**
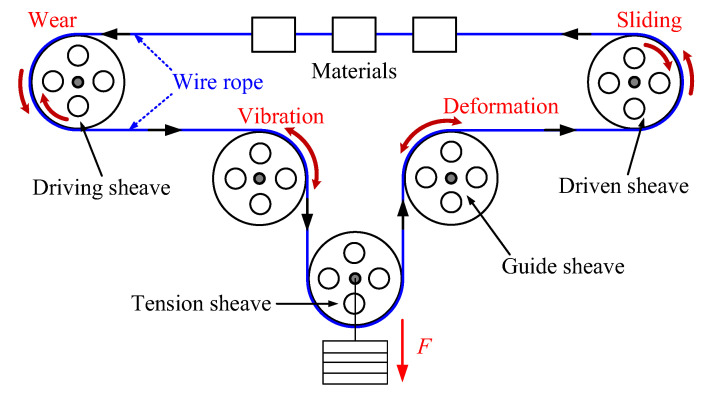
Schematic diagram of a wire rope-sheave transmission system.

**Figure 2 materials-15-07092-f002:**
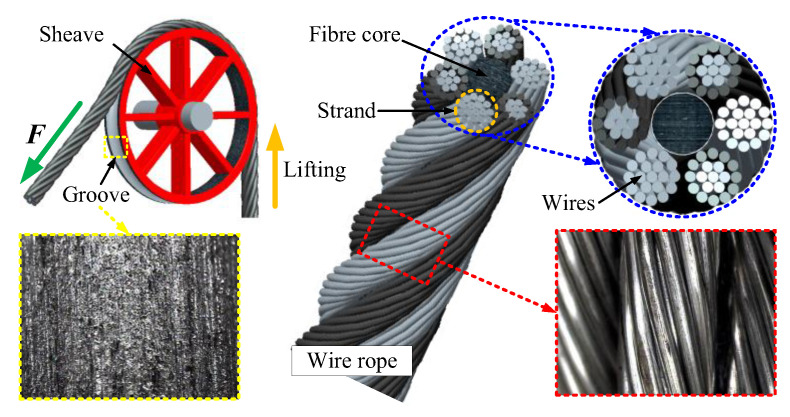
Surface structure of wire rope and sheave for friction and wear test.

**Figure 3 materials-15-07092-f003:**
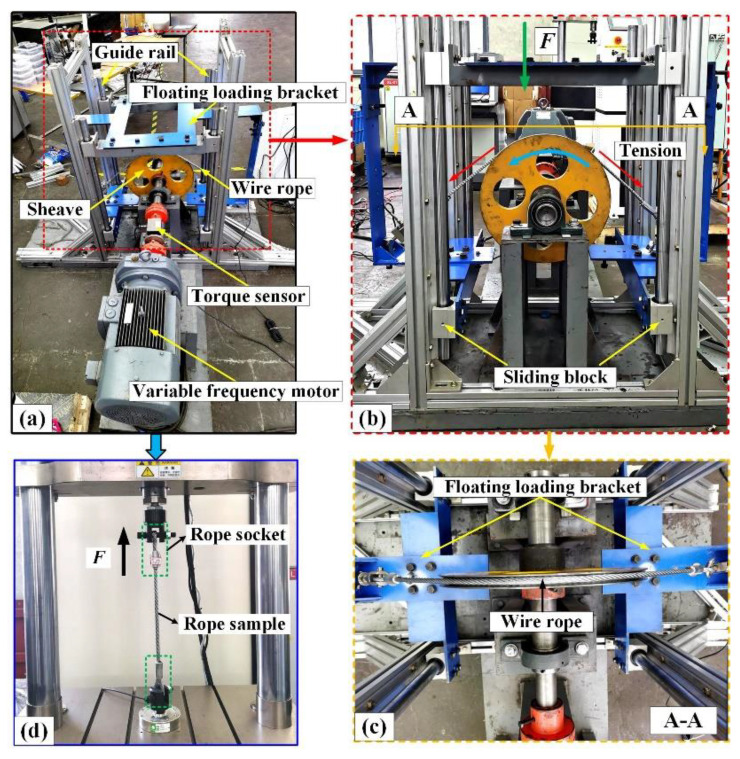
Structure of test rigs: (**a**) sliding friction and wear test rig of wire rope and sheave; (**b**) and (**c**) front and plan view of the friction and wear tester; (**d**) breaking tensile testing machine of wire rope.

**Figure 4 materials-15-07092-f004:**
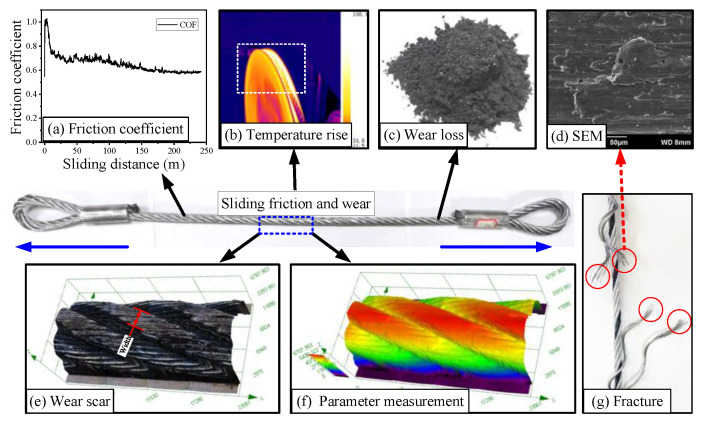
Main research parameters of the tribological behavior of the transmission wire rope.

**Figure 5 materials-15-07092-f005:**
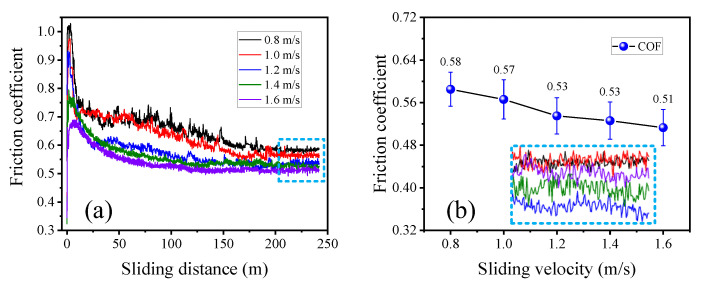
COF curves between the wire rope and the sheave under different sliding conditions: (**a**) process variation curves with sliding distance; (**b**) variation of COF with sliding velocity.

**Figure 6 materials-15-07092-f006:**
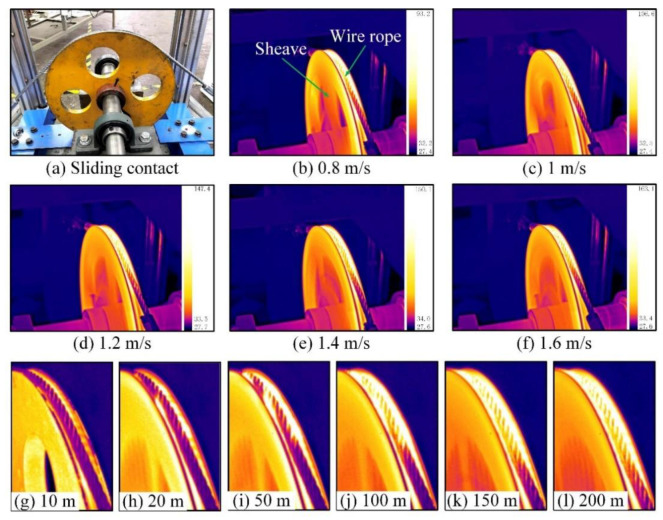
Infrared thermal images of sliding friction between wire rope and sheave: (**a**) physical map of rope-sheave contact; (**b**–**f**) infrared thermal image of wire rope under different sliding velocities; (**g**–**l**) infrared thermal image of the wire rope under different sliding distances.

**Figure 7 materials-15-07092-f007:**
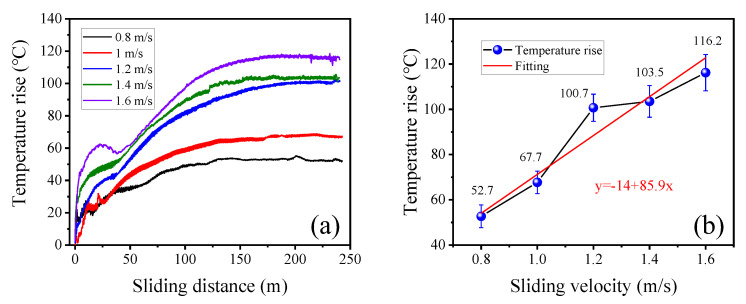
Variation curves of the maximum friction temperature increase in the contact region between wire rope and sheave under different sliding velocities: (**a**) process variation curves with sliding distance; (**b**) average temperature increase in the relatively stable stage.

**Figure 8 materials-15-07092-f008:**
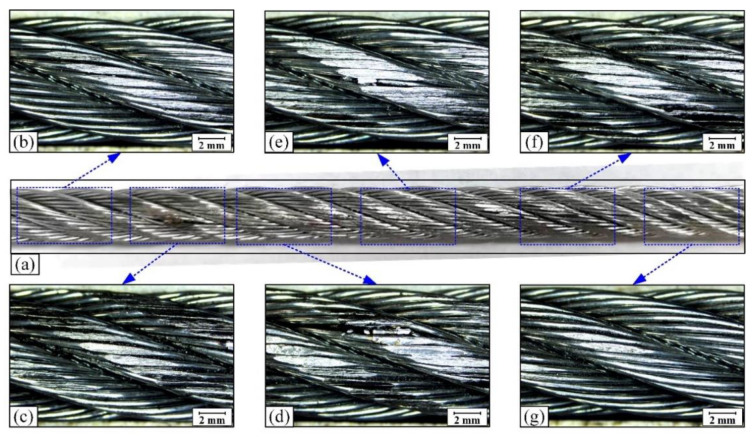
Optical micrographs of worn surface of wire rope at different positions. (**a**) wear scar on rope sample; (**b**−**d**) wear surface on the left side of the wear scar; (**e**−**g**) wear surface on the right side of the wear scar.

**Figure 9 materials-15-07092-f009:**
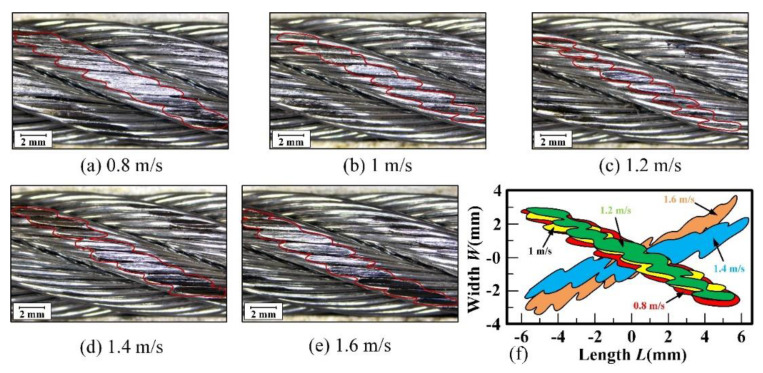
Distribution of wear scars on rope strand under different sliding velocities: (**a**−**e**) wear scars; (**f**) profile of wear scar under different sliding velocities.

**Figure 10 materials-15-07092-f010:**
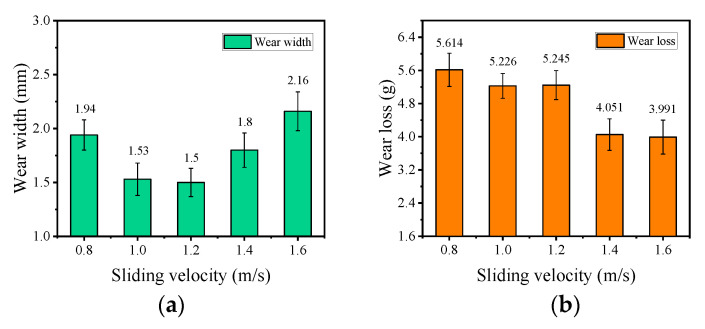
Wear characteristic parameters of wire rope under different sliding velocities: (**a**) maximum wear width; (**b**) weight loss.

**Figure 11 materials-15-07092-f011:**
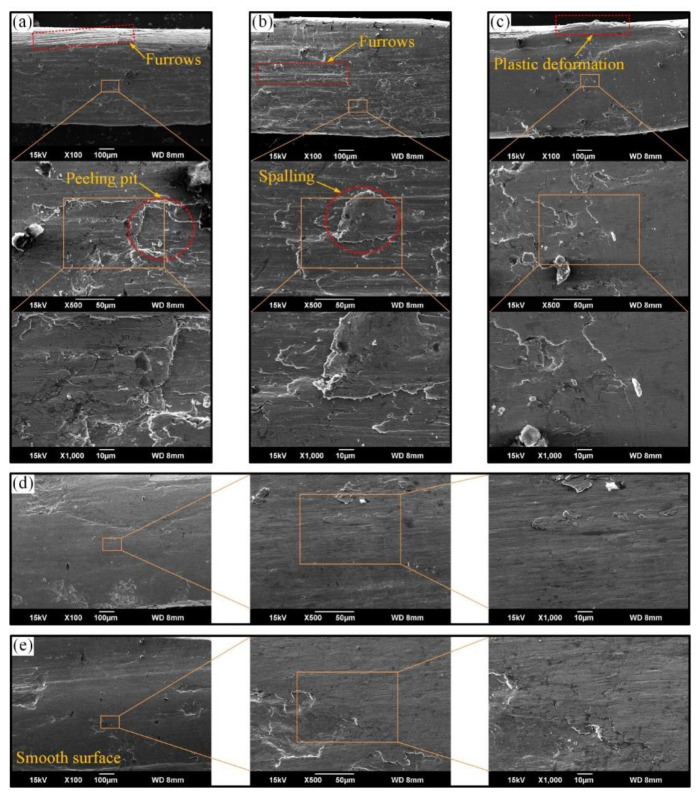
SEM of wear surface of rope wires under different sliding velocities: (**a**) 0.8 m/s; (**b**) 1 m/s; (**c**) 1.2 m/s; (**d**) 1.4 m/s; (**e**) 1.6 m/s.

**Figure 12 materials-15-07092-f012:**
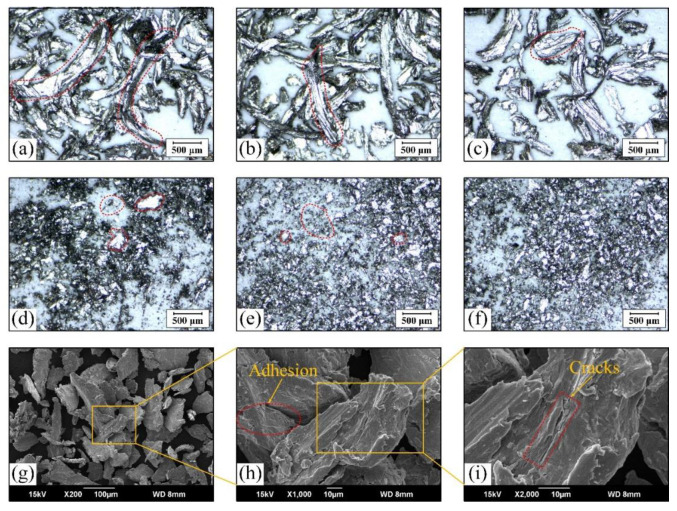
Wear debris: (**a**–**c**) large size debris; (**d**–**f**) small size debris; (**g**–**i**) morphology under SEM.

**Figure 13 materials-15-07092-f013:**
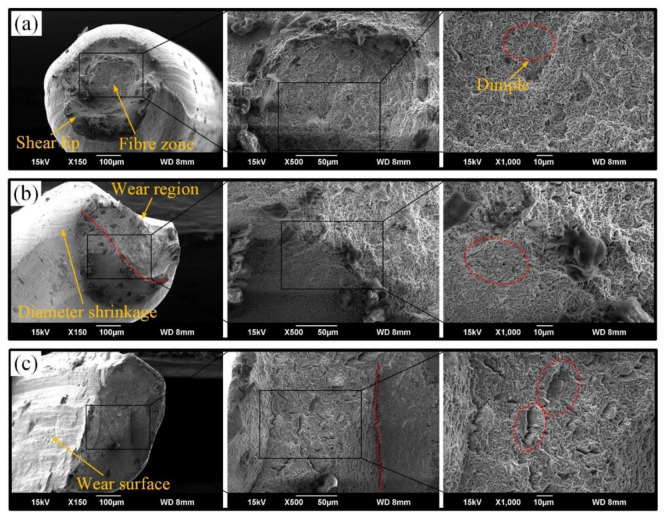
SEM of fracture surface of rope wires with different surface wear: (**a**) without surface wear; (**b**) mild wear; (**c**) severe wear.

**Table 1 materials-15-07092-t001:** Chemical composition of the steel wires.

Element	Fe	C	Si	Mn	Ni	S and P
Mass fraction (wt%)	98.69	0.87	0.39	0.03	0.02	<0.01

**Table 2 materials-15-07092-t002:** Structure parameters of the wire rope.

Parameter	Value
Length of the rope sample (mm)	600
Diameter of the rope (mm)	9.3
Radius of the steel wires (mm)	0.3
Strand lay length (mm)	70
Strand lay angle (°)	15.5
Strand lay direction	Right

**Table 3 materials-15-07092-t003:** Experimental parameters of the sliding friction and wear.

Parameters	Values
Sliding velocity (m/s)	0.8, 1, 1.2, 1.4, 1.6
Sliding distance (m)	240
Contact load (N)	600
Contact angle (°)	45
Contact arc length (mm)	157
Ambient temperature (°C)	27 ± 5
Relative humidity (%)	60 ± 10
Atmosphere	Laboratory air

## Data Availability

The data presented in this study are available on request from the corresponding author.

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
