# Peer review of "Sliding Friction and Wear Characteristics of Wire Rope Contact with Sheave under Long-Distance Transmission Conditions"

_materials, 2022, doi:10.3390/ma15207092_

Round 1

Reviewer 1 Report

The paper is overall very well written, although the aim/purpose of the study seems not so significant, or at least not clearly described in the introduction. I agree with the authors that there will be sliding wear between the rope and sheave, but it is still an rolling/sliding contact and the rolling part dominates the whole process. The significance of the sliding part should be emphasized in some way. Just saying that the sliding wear has not been studied before is not enough. I think this part is the main area which could be improved. Otherwise, it is a good paper and deserves to be published in the journal.

Author Response

Comments:

The paper is overall very well written, although the aim/purpose of the study seems not so significant, or at least not clearly described in the introduction. I agree with the authors that there will be sliding wear between the rope and sheave, but it is still an rolling/sliding contact and the rolling part dominates the whole process. The significance of the sliding part should be emphasized in some way. Just saying that the sliding wear has not been studied before is not enough. I think this part is the main area which could be improved. Otherwise, it is a good paper and deserves to be published in the journal.

Response: Thank you for your advice. In the introduction, we emphasized the significance of studying the sliding friction behavior between the wire rope and the sheave, as shown in line 171-178 (Page 3).

“Although the rolling contact is the main contact form between the wire rope and the sheave during the operation of the transmission system, the friction and wear caused by the sliding contact is more serious to the wire rope. Moreover, the use frequency of the wire rope is high, and the wear is a process of continuous accumulation and deterioration during long-term service. Therefore, it is of great engineering significance to study the sliding friction characteristics between the wire rope and the sheave to en-sure the safe service of wire ropes and reduce the damage.”

Reviewer 2 Report

Despite the efforts done by the authors, and the manuscript contains a good scientific story, there are some important points that should be considered by the authors.

·       Abstract

1.     The authors didn't mention the material of the wire rope or the sheave.

2.     The authors should write the different sliding speeds used and mention the observed wear mechanisms.

·       Introduction

1.     Introduction of manuscript is long and contains repetitive general sentences and needs revision. In addition, the aim of the manuscript is more like a summary of the experimental section, preferably paraphrasing.

·       Materials and Methods

1.     Page 3, Lines 136-137, it is better to present the wire material chemical composition in a table.

2.     Page 4, Line 140, the author use material name “Q235” according to the Chinese standard, it is preferable to write the equivalent material in an international well known standard “DIN EN 235JR” or “ASTM A36”.

3.     Page 4, Line 149, “local view” should be changed to “front and plan view”.

4.     Page 6, Line 183, low accuracy electronic balance used “0.1”.

·       Results and Discussion

1.     Author didn’t mention the results of the tensile test conducted.

2.     Page 6, Lines 191-192, “different sliding velocities are presented in Figure 5” should be changed to “different sliding distance and sliding speeds are shown in Fig. 5(a,b) respectively”.

3.     Page 6, Line 195, “(b) average COF in the relatively stable stage” should be changed to “(b) variation of COF with Sliding velocity”.

4.     Page 7, Line 225. Please check “The friction resistance of the wire rope under high-speed sliding conditions”. I think should be deleted.

5.     Page 8, form Lines 255 to 267, This part needs to be rewritten and use the language of numbers to clarify the meaning

6.     Page 10, Line 346, sliding speed “(N)” should be changed to “(m/s)”.

7.     Page 10, Line 348, “wear loss” should be replaced be “weight loss”.

8.     Page 11, Figure 11, should be modified ( color of lines, writing on the worn surface,…to be more clear

Author Response

Thank you for your comments. We have studied the comments carefully and have made revisions and modifications which we hope meet with approval. Additionally, the revised portions are marked up using “Track Changes” in revised manuscript. The main corrections in the paper and the responds to the comments are as follows:

  • Abstract
    1. The authors didn't mention the material of the wire rope or the sheave.

Response: The material of the wire rope and the sheave was added. As shown in line 17 (Page 1).

  1. The authors should write the different sliding speeds used and mention the observed wear mechanisms.

Response: The velocity parameters were added in line 16. The wear mechanisms were added in line 25-26 (Page 1).

  • Introduction
    1. Introduction of manuscript is long and contains repetitive general sentences and needs revision. In addition, the aim of the manuscript is more like a summary of the experimental section, preferably paraphrasing.

Response: Thank you for your comments, we have carefully checked and carefully modified the introduction. The similar and repetitive descriptions were deleted. Some sentences have been rewritten and adjusted. The detailed modification information has been marked in the revised manuscript.

  • Materials and Methods
  1. Page 3, Lines 136-137, it is better to present the wire material chemical composition in a table.

Response: the chemical composition of the wire material has been presented in Table 1, as shown in line 196 and 198 (Page 3).

  1. Page 4, Line 140, the author use material name “Q235” according to the Chinese standard, it is preferable to write the equivalent material in an international well known standard “DIN EN 235JR” or “ASTM A36”.

Response: The material name “Q235” have been changed to “ASTM A36”, as shown in line 226 (Page 4). Thank you for your advice.

  1. Page 4, Line 149, “local view” should be changed to “front and plan view”.

Response: We have changed it, as shown in line 235 (Page 4). Thank you for your advice.

  1. Page 6, Line 183, low accuracy electronic balance used “0.1”.

Response: The balance used in this paper is high-precision analytical balance. The measurement accuracy of the analytical balance is 0.1 mg. We have made a supplementary explanation in line 284-285 (Page 6).

  • Results and Discussion
  1. Author didn’t mention the results of the tensile test conducted.

Response: The purpose of breaking tensile test is to analyze the influence of surface wear on the fracture characteristics of the wire rope. The test only provides a tensile load for the worn rope sample. Therefore, based on the test results, we only observed and analyzed the fracture morphology of the broken wires. The results are shown in Figure 13, (Line 537, Page 13). Additionally, a description of the tensile test is supplemented in line 264-265 (Page 5).

  1. Page 6, Lines 191-192, “different sliding velocities are presented in Figure 5” should be changed to “different sliding distance and sliding speeds are shown in Fig. 5(a,b) respectively”.

Response: We have changed it, as shown in line 292-293 (Page 6). Thank you for your advice.

  1. Page 6, Line 195, “(b) average COF in the relatively stable stage” should be changed to “(b) variation of COF with sliding velocity”.

Response: We have changed it, as shown in line 296 (Page 6). Thank you for your advice.

  1. Page 7, Line 225. Please check “The friction resistance of the wire rope under high-speed sliding conditions”. I think should be deleted.

Response: We have deleted it, as shown in line 332 (Page 7). Thank you for your advice.

  1. Page 8, form Lines 255 to 267, This part needs to be rewritten and use the language of numbers to clarify the meaning.

Response: This section is a description and analysis based on the results in Figure 6. By comparing the changes of infrared thermal images of wire rope under different sliding distances, the variation law of temperature is analyzed. Based on the image can only analyze the changes in temperature distribution and qualitative comparison. It is different to described the result using the language of numbers. Moreover, the quantitative analysis results can be obtained from Figure 7, which have been presented in the paper, as shown in line 375-407 (Page 8 and 9).

  1. Page 10, Line 346, sliding speed “(N)” should be changed to “(m/s)”.

Response: We have changed it, as shown in line 454 (Page 10). Thank you.

  1. Page 10, Line 348, “wear loss” should be replaced be “weight loss”.

Response: We have changed it, as shown in line 456 (Page 10). Thank you for your advice.

  1. Page 11, Figure 11, should be modified ( color of lines, writing on the worn surface,…to be more clear

Response: We have modified the line, text and the color of Figure 11 (line 483, Page 11), which make the picture look clearer. Thank you for your advice.

Reviewer 3 Report

I have read the manuscript titled " Sliding Friction and Wear Characteristics of Wire Rope Contact with Sheave under Long-distance Transmission Conditions" for possible publication in the journal Materials

My review report is attached, and I recommend minor revision for this submission

Author Response

Thank you for your comments. We have studied the comments carefully and have made revisions and modifications which we hope meet with approval. Additionally, the revised portions are marked up using “Track Changes” in revised manuscript. The main corrections in the paper and the responds to the comments are as follows:

Point 1: Keywords can be introduced more based upon the research

Response: We have added three new keywords: sheave; friction heat; wear debris. Line 28-29 (Page 1).

Point 2: Introduction should be reduced. The novelty should be clearly defined at the end of the introduction.

Response: We deleted some descriptions and adjusted some sentences. The introduction was shortened. The detailed modifications were marked in the revised manuscript. Additionally, the end of this section was rewritten, which makes the novelty of this paper clearer, as shown in line 171-184 (Page 3).

Point 3: How were these parameters fixed? How many trials of the experiment were done?

Response: The parameters of the friction test are determined based on the service conditions of the transmission wire rope, because the contact parameters between the wire rope and the pulley vary greatly under different operating conditions. The parameters selected in this article are within the range of change, so they are relatively reasonable. The related descriptions “In engineering applications, the types of sheaves used in conjunction with wire ropes are diverse, and the contact angle between the wire rope and the sheave varies from 0⁰ to 180⁰. Additionally, the sliding speed and contact load between the wire rope and the sheave are constantly changing during operation. Therefore, this paper chooses the sliding velocity as the research variable, and the selected test parameters are real under service conditions.” were added in line 254-259 (Page 5).

The friction experiment was repeated three times, as shown in line 252-253 (Page 5).

Point 4: What was the load given as the tensile load?

Can you give the values of the tensile strength, YS and ductility? Please support with the stress strain graph.

Response: The purpose of breaking tensile test is to analyze the influence of surface wear on the fracture characteristics of the wire rope. The test only provides a tensile load for the worn rope sample. Therefore, based on the test results, we only observed and analyzed the fracture morphology of the broken wires. And we did not record the relevant mechanical parameters of the wire rope during the tensile test. To make readers understand the content of this article more accurately, we have modified the description of the tensile test, as shown in line 262-265 (Page 5).

Point 5: How many trials of experiment were done? (Section 2.3 Test parameters and methods)

Response: Each test result was measured three times. It was added in line 285 (Page 6).
